# Natural Vibration Characteristics Analysis of a High-Rise Reinforced Masonry Structure Based on Field Test Data

**Baofeng Zhou** [1,2]**, Bo Liu** [3]**, Xiaomin Wang** [1,2,*]**, Jingchang Kong** [3]  **and Cong Zhang** [1,2]

1    Key Laboratory of Earthquake Engineering and Engineering Vibration, Institute of Engineering Mechanics, China Earthquake Administration, Harbin 150080, China
2    Key Laboratory of Earthquake Disaster Mitigation, Ministry of Emergency Management, Harbin 150080, China
3    School of Civil Engineering, Yantai University, Yantai 264005, China
\*    Correspondence: xiaomin_wang@iem.ac.cn

**Abstract:** In structural response array observation, the vibration response of a structure during an earthquake or from the natural environment is recorded and stored using high-sensitivity strong motion seismographs, and the dynamic characteristics of the structure are analyzed and determined using random signal data processing technology. Due to the use of field test data for analysis, this may be the most accurate and effective way to obtain the actual characteristics of the structure, which can be further used to verify the accuracy of theoretical analysis, experimental results, and numerical simulations. Therefore, this technique plays an important role in earthquake prevention and disaster reduction, with the application of strong motion observation data. In this paper, field vibration tests were performed on the highest reinforced masonry structure in China. With the test environmental vibration data, the natural vibration frequency values and mode shapes of the structure were identified using the peak picking method. A numerical modal analysis was then performed to verify the accuracy of the field test results. In addition, the structural response records obtained during an earthquakes in Songyuan were also used to identify the natural vibration frequency of the structure and the changes in the natural vibration frequency before, during, and after the earthquake. The results showed that the structure was not damaged during the earthquake and remained in an elastic state.

**Keywords:** reinforced masonry structure; structural response array; the peak picking method; field vibration tests; natural vibration frequency; mode shapes

## 1. Introduction

With the rapid development of society and the economy, earthquake protection and disaster reduction has become more urgent. In recent years, several strong earthquakes have caused a large number of casualties and huge economic losses [1]. In order to reduce earthquake damage, national and regional governments have increasingly realized that the necessary investments are of great significance for strong motion observation and the research of structures [2,3]. This aims to apply strong motion seismographs to observe ground motion or the response of engineering structures during earthquakes or with environmental vibration, to provide basic research data for structural engineering, and to further study the seismic performance of structures, so as to minimize the loss of lives and property [4–6].

In 1932, the first strong motion accelerometer was successfully developed in the United States, and the first group of strong motion acceleration records was obtained during the Long Beach earthquake in the following year. Since then, strong motion observation has gradually attracted a great deal of attention in related fields. The ANSS plan, proposed by the USGS in 1999, deployed 6000 strong motion seismographs in specific areas of 26 related cities [7]. In Japan, 1400 strong motion seismographs had been installed by the 1980s.

After the Hanshin earthquake in 1995, construction plans for the K-NET, KiK-net, and JMA observation networks were successively launched [8]. By 1997, a total of 5855 strong motion seismographs had been installed in Japan [9]. In China, strong motion observation began in the 1960s, and strong motion observation arrays were arranged on a few structures. Due to the relatively undeveloped level of instruments and the limited number of arrangements, the strong motion records obtained were quite spars [1]. In the 1970s, Taiwan district began to carry out strong motion observation, and about 1500 strong motion instruments were set up throughout the whole island. In the Chi-Chi earthquake, a large number of strong motion records were obtained, which promoted the further development of strong motion seismology [9]. Considering economic factors, Iwan [10] pointed out that future strong motion arrays should be deployed in earthquake-prone places around the world, and that through mutual cooperation among countries, this can provide the greatest potential benefits from a large investment in strong motion observation.

At present, the proportion of strong motion observation arrays placed on engineering structures is not large, thus the available seismic response observation data from structural arrays are relatively few. In practical research, theoretical analyses, numerical simulations, and experimental tests are used to analyze the natural vibration characteristics and dynamic response of structures, and many new analysis methods have been proposed for structural seismic assessment [11–16]. All these analyses make different degrees of simplified assumptions about the used models, and the reliability of the models cannot be verified, due to a lack of response data of structures during strong earthquakes. In this case, the construction of structural response arrays and the considered application and analysis of existing strong motion data and environmental vibration data has become an effective way to promote the development of structural seismic engineering [17–20].

Combined with the environmental vibration data of a structural response array, Gallipoli et al. [21] analyzed the soil–structure interactions; Facke et al. [22] accurately estimated the natural frequency of a church in Germany. Ueshima et al. [23] carried out long-term health monitoring of an arch dam structure, and according to a frequency spectrum analysis of the main shock in the 2011 Northeast Pacific Earthquake and the environmental vibration records, it was found that the predominant frequency during the main shock was significantly decreased, and the frequency was generally recovered after the main shock. Astorga et al. [24] identified the natural frequency of a building through environmental vibration data and observed that the resonance frequency of the building continued to decrease until it reached a stable value, which showed the process of continuous damage of the structure under a cyclic dynamic load. Luo et al. [25] identified the natural frequency of a reinforced concrete structure using test environmental vibration data, extended the application of the spectral ratio method to the structure, and analyzed the seismic response of the structure, to verify the feasibility of the method; Wang et al. [26] identified the natural vibration characteristics of a building, based on test environmental vibration data, and according to the identification results, arranged the seismic response array on the floors with a large amplitude. Combined with the strong motion observation data of a structural response array, Celebi [27] identified the natural vibration characteristics of the Atwood Building through the seismic response records observed by the comprehensive earthquake monitoring network, and judged the possible occurrence of a resonance phenomenon, according to the proximity of the site frequency to the second-order natural frequency of the structure. Miyakoshi et al. [28] analyzed observation data of strong vibrations and found that the predominant period of Osaka Basin may have caused the resonance that occurred in high-rise buildings between 150 and 300 m high near Osaka City. Chandramohan et al. [29] analyzed the seismic response records of two buildings during the Kaikoura earthquake and pointed out that the structure experienced a significant ground motion amplification effect in 1–2 s, due to the site effect, which should be considered critically in seismic fortification. Wu et al. [30] demonstrated an effective application of seismic environmental noise interferometry in structural health monitoring and risk assessment through an environmental noise analysis of the Shanghai Tower.

In general, the processing methods of array observation data and the application of data in identifying the natural vibration characteristics of structures are still in their infancy, and the application of array observation data requires further research. With the development of real-time data processing methods, the demand for structural response arrays is increasing progressively, and this development calls for constant investment. Therefore, this is particularly important for the analysis of structural response array data, especially for special types of structure. In this paper, the highest reinforced masonry structure in China was selected, to identify the natural vibration frequency and mode shape of the structure from environmental vibration tests and numerical analysis. At the same time, a group of structural response records captured during the Songyuan earthquake were processed and analyzed, and the health status of the structure was judged by the changes of natural vibration frequency information of the structure before, during, and after the earthquake.

## 2. Information about the Reinforced Masonry Structure

In this paper, a field test of a reinforced masonry structure was carried out using the environmental excitation method. The structure is located in Songbei District, Harbin, China, and the main structure is 98.8 m high, which is the highest reinforced masonry structure in China. In fact, it exceeds the maximum applicable height specified in "Code for Design of Masonry Structures" (GB50003-2001) [31] and "Code for Seismic Design of Buildings" (GB50011-2010) [32]. The plan of a typical floor and front elevation for the structure are shown in Figures 1 and 2. The structure is composed of one basement floor and 29 floors above ground, and the above ground part is composed of 28 ordinary floors and an elevator floor. The heights of the first, second, 3–27th, 28th floor, and the elevator floor are 4.5 m, 4.2 m, 3.4 m, 4.5 m, and 4.5 m, respectively. The total height of the structure is 101.8 m, which is an ultra-high building. It was the first high-rise reinforced concrete block masonry structure with a height of over 100 m in China. The structure adopts the form of a block casting shear wall, the structure size is similar to a rectangle, and the plane size is about 35.1 m × 16.2 m. The total construction area is 16,187 m$^2$. The thickness of the outer walls of the 1–14th floors is 290 mm. The outer walls of the 15–28th floors and the elevator room are 190 mm thick. The height to width ratio of the main structure is 6.59. The construction site is classified as class III, and the earthquake design is in the first group. The seismic precautionary intensity is 6 degrees, i.e., the peak acceleration of ground motion is 0.05 g. The characteristic period is 0.45 s.

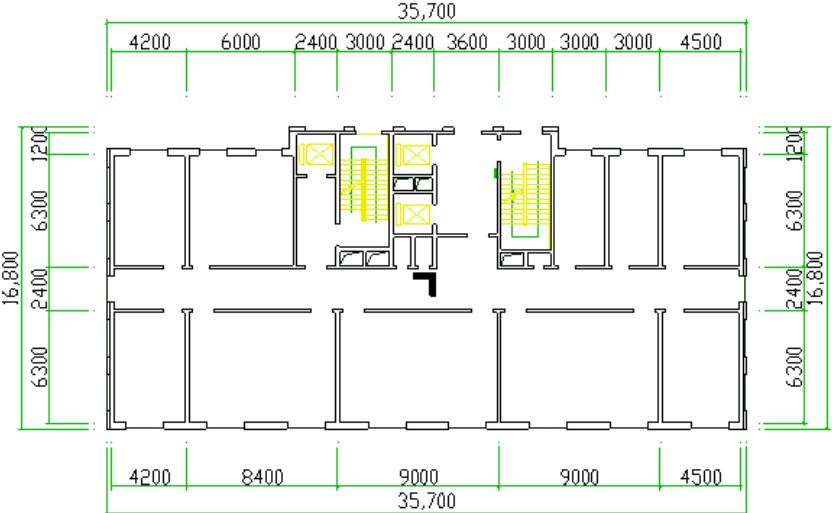

**Figure 1.** Plan of a typical floor (The black block is the sensor position; unit: mm).

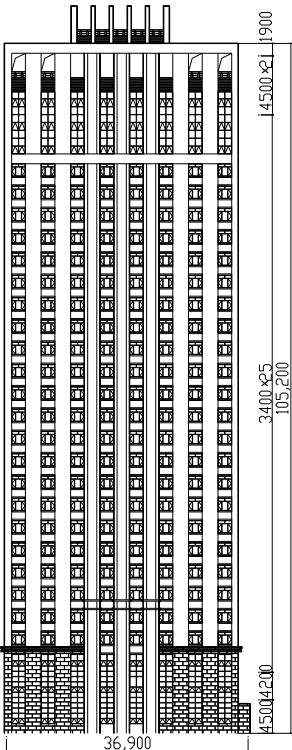

**Figure 2.** Front elevation.

The structure was formed with a block pouring wall structure. In terms of construction technology, for the 190-mm thick wall, a 190-mm block with strength of MU20 and the hole rate of 45% was adopted. For the 290-mm thick wall, a new 290 mm block with a hole rate of 53% was adopted. The strength grade of mortar was Mb20, and the strength grade of the core concrete was C40. The masonry was 100% fully core-filled, according to the code for design of masonry structures (GB50003-2010) [31]; for the 290-mm-thick wall and 190-mm-thick wall, the compressive strength is 12.37 Mpa and 11.45 Mpa, respectively; the modulus of elasticity is 24,740 Mpa and 22,900 Mpa, respectively. In addition, the concrete C30 and steel HRB400 were used for ring beams, floor slab, connecting beams, and the beams under the floor slab of the masonry shear wall.

## 3. Structural Dynamic Characteristics Identification Method

### 3.1. Environmental Excitation Method

The environmental excitation method aims to infer the dynamic characteristics of a structure by recording the vibration response of the target structure in the environment. This tiny pulsed vibration of the structure is caused by a variety of factors, such as crustal activity, vehicle movement, machine operation, and wind vibrations. During the test, there is no need for special excitation equipment, so it is not limited by the structural form and size [33]. The analysis of structural dynamic characteristics based on environmental vibration testing is a hot issue in the field of structural engineering, and its area of application includes the defense industry, transportation engineering, civil engineering, mechanical engineering, and aerospace [34,35]. At present, along with the rapid development and continuous improvement of testing technology and finite element theory, it is a key issue that needs attention in the future for accurate and real-time judgment of damage location and damage degree. In addition, for a specific structure, the difference of frequency characteristics of the structural environmental vibration data before and after an earthquake is able to reflect the change of the structural characteristics and enable judging the damage to the structure [36,37].

A flow chart of the environmental excitation method is shown in Figure 3. A tiny vibration response signal is collected by the sensor, converted into an electrical signal and amplified, and then transmitted to the collecting instrument for automatic recording and storage; then the data are processed by the modal identification method, to obtain the natural vibration frequency, the structural mode shape, and other dynamic characteristics.

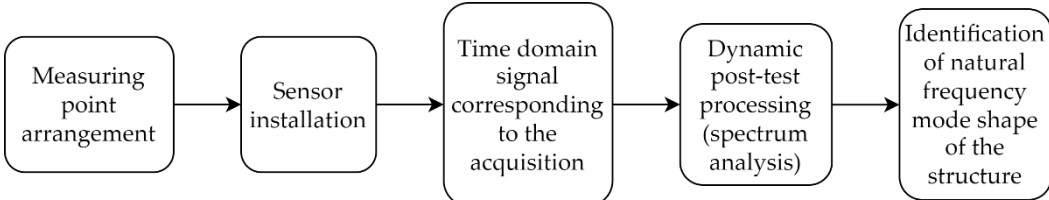

**Figure 3.** Environmental excitation method flow chart.

Currently, the modal identification methods commonly used include frequency domain methods, such as peak picking, frequency domain decomposition, and enhanced frequency domain decomposition, as well as time domain methods, such as random decrement, random subspace algorithms, and time series [38–41]. In this paper, the peak picking method was adopted for the modal identification of the structure.

### 3.2. Data Processing Method

In order to ensure the reliability of the research results, preprocessing of raw data, which includes baseline processing, digital filtering, Fourier transform, and data smoothing, is needed, to prevent external conditions from interfering with the original data, and at the same time, to enable the seismic response observation records to truly reflect the vibration characteristics of the structure.

For baseline processing, based on the least square method, a linear fitting method is adopted to generate a new linear function for the original discrete points, and each value of the raw data is used to subtract the value of each corresponding point of the new linear function, so as to achieve the effect of baseline initialization of the raw data. In this paper, a Butterworth filter was used to perform band-pass filtering on the data after baseline processing, to obtain the ground motion records diminishing the influence of high and low frequencies; and the frequency band range was set to 0.1–35 Hz. On this basis, Fourier transform was performed on each section of data, to obtain the Fourier amplitude spectrum. Finally, the Konno–Ohmachi method [42] was used for smoothing, to obtain the smoothed Fourier amplitude spectrum.

### 3.3. Peak-Picking Method

The peak-picking (PP) method [38] uses input and output responses to obtain the frequency response function of the structure, and according to the principle that the frequency response function will have a peak near the natural frequency of the structure, the characteristic frequency is determined using the peak value on the power spectral density curve [43]. After determining the natural frequency, the mode shape of the structure can be determined by the ratio of amplitudes at the natural frequency for different measuring points.

The frequency response function of the structure can be written as:

$$H(j\omega) = \sum_{i=1}^{n} \frac{1}{j\omega - \lambda_i} \phi_i l_i^T \tag{1}$$

where $\omega$ is the natural frequency of the structure; $\phi_i \in R^m$ represents the modal vector of the structure; $j$ is the number of output responses; $n$ represents the modal order of the structure; $\lambda_i$ corresponds to the eigenvalue of the continuous time state model of the

structure; $l$ is the modal participation coefficient vector. The $i$th-order frequency $\omega_i$ and damping ratio $\zeta_i$ of the corresponding structure are:

$$\lambda_i, \overline{\lambda}_i = -\zeta_i \omega_i \pm j\sqrt{1 - \zeta_i^2}\omega_i \phi_i \in R^m \tag{2}$$

In the case of low damping, the eigenvalue $\lambda_i \approx -\zeta_i \omega_i + j\omega_i$ of the continuous state space model of the corresponding structure can be substituted into Equation (1) to obtain the maximum value of the frequency response function corresponding to the natural frequency $\omega_i$ of the $i$th order, as follows:

$$H(j\omega_i) = \frac{1}{\zeta_i \omega_i}\phi_i l_i^T \tag{3}$$

Therefore, the natural frequency of the structure can be obtained by picking up the peak value of the frequency response function. For the modal parameter identification method using environmental excitation, it is usually impossible to obtain the input response of the structure. Therefore, the Fourier spectrum of the output response signal is usually used to replace the frequency response function, to identify the modal parameters of the structure.

The peak-picking method is the simplest method for identifying modal parameters, and it is suitable for low damping ratio structures with relatively separated modes. It has a fast recognition speed; in the case of a not very dense modal, it also has a good recognition accuracy; thus, it is widely used in construction engineering.

## 4. Environmental Vibration Test Data Analysis

In order to meet the requirements of environmental vibration testing and prevent interference from the external environment, the field test was performed at night [44]. The main equipment for the field test included acceleration sensors, digital recorders, and laptops. The acceleration sensor adopted a triaxal EpiSensor force balance accelerometer, FBA ES-T sensor; its dynamic range is 155 dB+, the bandwidth is DC to 200 Hz, and the full scale can be set within the range of ±0.25 g to ±4 g. The recorder was a Basalt, which is a 4 sensor channel digital recorder with an external Episensor Triaxial Deck, and each channel is a 24-bit Delta Sigma converter. It supports multiple sample rates, data formats, and telemetry protocols. The test direction included the long-axial translation and short-axial translation direction of the structure. The field tests were carried out from the basement floor to the top floor. The acceleration sensor on each floor was located at the center of the plane, as shown in Figure 1. Before the test, the accelerometer was first calibrated, to ensure the normal operation of the instrument. In the test, the data sampling frequency was 200 sps, and the collection time was longer than 10 min, to ensure the stability and objectivity of the sampled data.

### 4.1. Environmental Vibration Data Preprocessing

During environmental vibration data preprocessing, each record in the original data is stored for a long time, and has more than 40,000 sampling points. In order to facilitate data analysis, each piece of data was divided into 10 segment records, which were intercepted with a length of 4000 sampling points. Each segment of data was preprocessed by the method described above. The acceleration time history curves of part of the original data, before and after baseline initialization and band-pass filtering, are shown in Figure 4. The Fourier amplitude spectra of two segments of data, with and without smoothing, for a representative floor are shown in Figure 5. The black line in the figure is the Fourier amplitude spectrum curve of the original data, and the red line is the Fourier amplitude spectrum curve of the processed data. It is not difficult to see the smoothing effect of the original environmental vibration data using the above method; the sharp part of the original data is removed, and at the same time, the spectrum characteristics of the original data in each frequency band are well preserved.

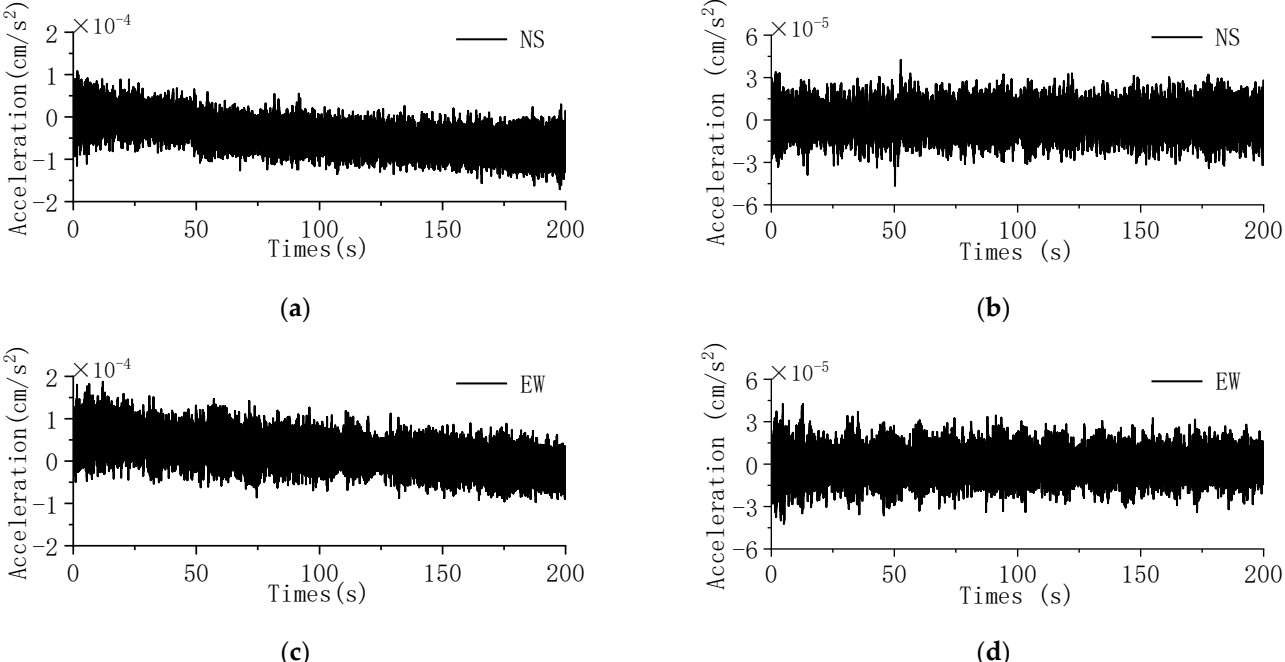

**Figure 4.** Acceleration time history curves of parts of the original data (**a**,**c**) before and (**b**,**d**) after baseline initialization and band-pass filtering.

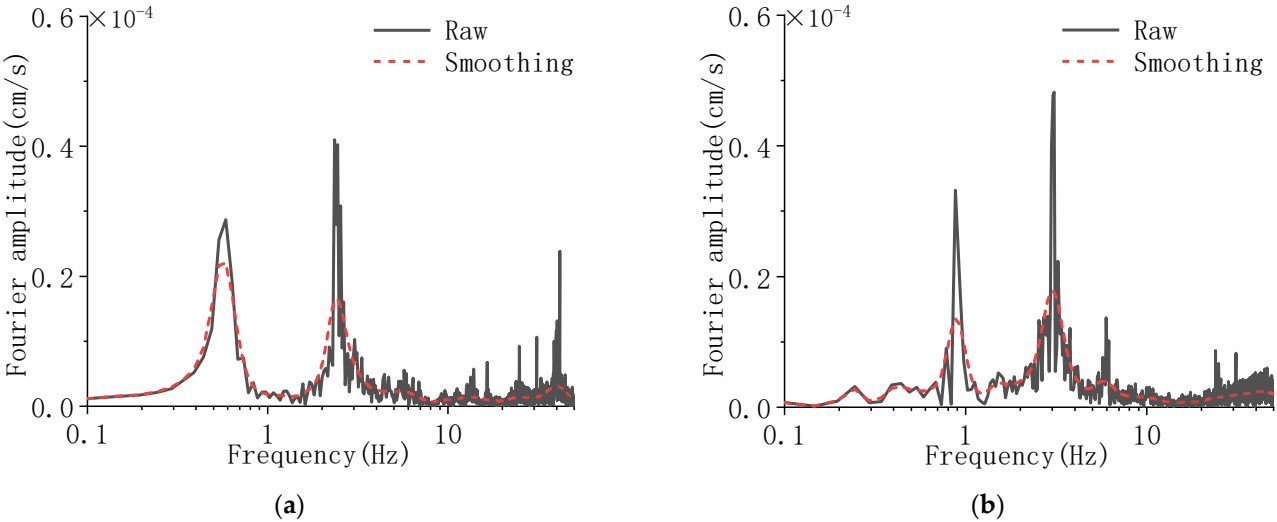

**Figure 5.** Fourier amplitude spectra of several segments of data, with and without smoothing (**a**) NS; (**b**) EW.

### 4.2. Results Analysis

The test environmental vibration data are not presented in this section for the sake of brevity. However, from the environmental vibration data of different floors, it can be concluded that with an increase of height, the vibration response of the structure became gradually larger.

The peak-picking method was used to identify the first three-order natural vibration frequencies of the structure in the NS and EW directions. The Fourier amplitude spectra of the 10 segments of data for several representative floors in two directions are shown in Figure 6.

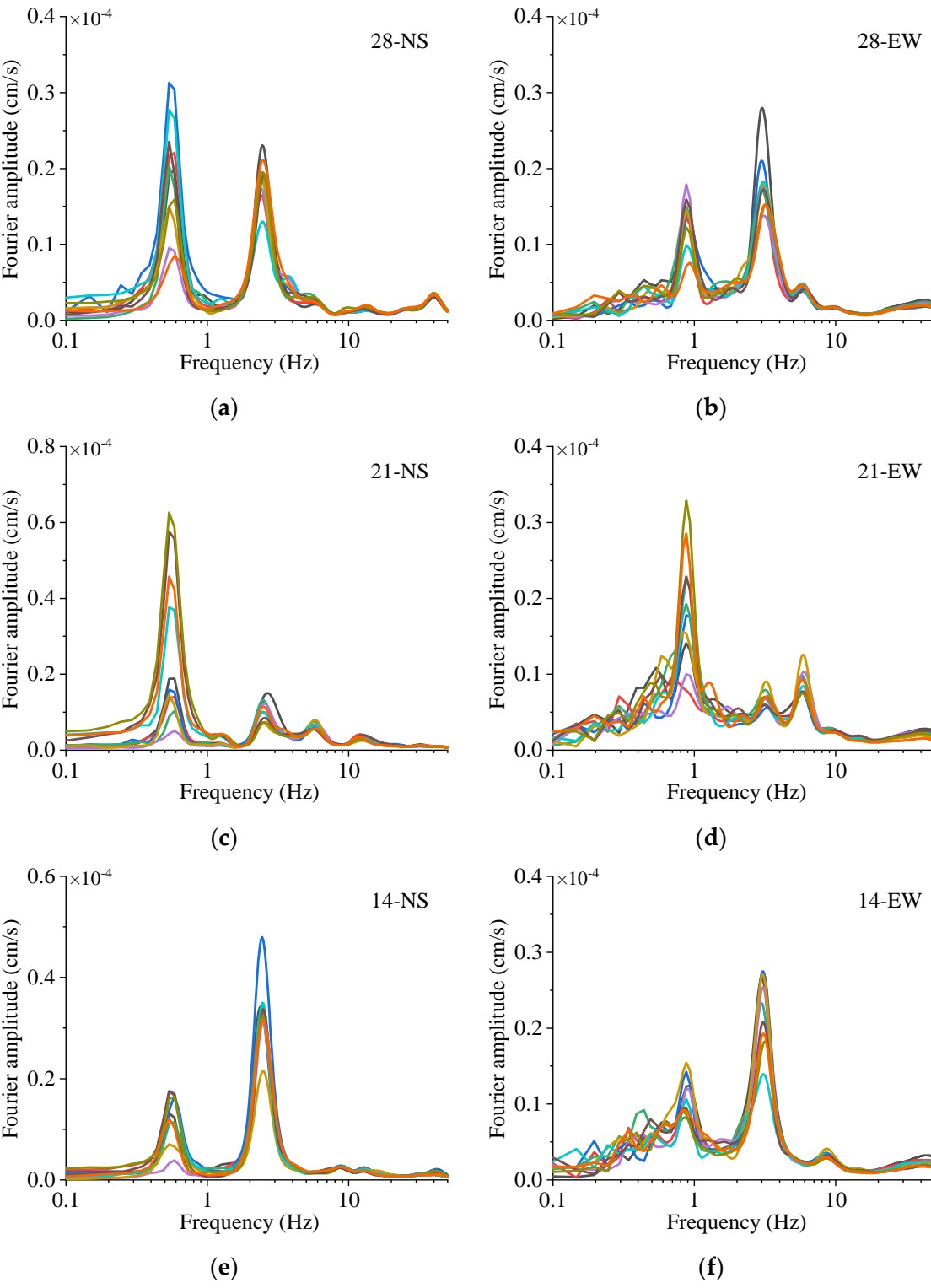

**Figure 6.** *Cont.*

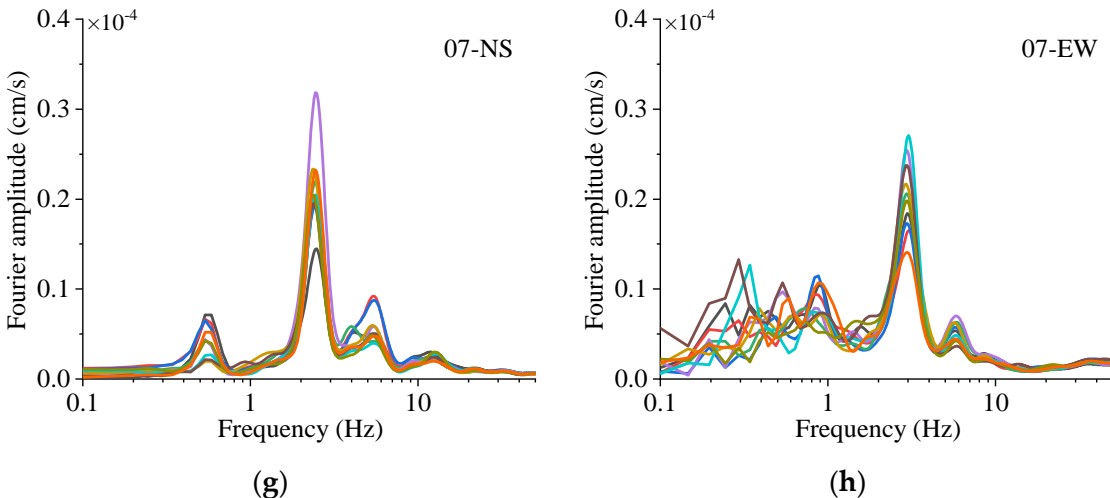

**Figure 6.** Fourier amplitude spectra of 10 segments of data with smoothing for several representative floors (**a**) 28 floor in NS; (**b**) 28 floor in EW; (**c**) 21 floor in NS; (**d**) 21 floor in EW; (**e**) 14 floor in NS; (**f**) 14 floor in EW; (**g**) 7 floor in NS; (**h**) 7 floor in EW.

It can be seen that the Fourier amplitude spectra obtained from the 10 segments of data of the same floor and the data of different floors are consistent with the overall fluctuation trend. For the NS direction, the Fourier amplitude spectra of the floors below the 5th floor have some differences in the low frequency stage, while the Fourier amplitude spectra of other floors are relatively close. It can be seen that there are distinct peaks around the frequencies of 0.55 Hz, 2.46 Hz, and 5.50 Hz. For the EW direction, there is also a certain difference in the low-frequency stage of the Fourier amplitude spectra of the floors below the 5th floor, while the results of other floors are relatively close, and obvious peaks can be seen at the frequencies of 0.87 Hz, 3.05 Hz, and 5.77 Hz. The recognition of higher-order frequencies is not very accurate, and the error may be larger, so this was not identified in this paper. In addition, considering that the peak frequency below the 5th floor was unstable, the results of floors 5–28 were selected to obtain the first three-order frequencies in two directions, as listed in Table 1.

**Table 1.** The first three-order frequencies in two directions (Unit: Hz).

| Floor | First Order | | Second Order | | Third Order | |
|---|---|---|---|---|---|---|
| | NS | EW | NS | EW | NS | EW |
| 28 | 0.557 | 0.884 | 2.441 | 3.071 | 5.447 | 5.796 |
| 27 | 0.557 | 0.884 | 2.437 | 3.052 | 5.469 | 5.743 |
| 26 | 0.557 | 0.884 | 2.427 | 2.969 | 5.127 | 5.859 |
| 25 | 0.548 | 0.874 | 2.412 | 2.939 | 5.532 | 5.713 |
| 24 | 0.552 | 0.884 | 2.349 | 2.861 | 5.542 | 5.645 |
| 23 | 0.548 | 0.884 | 2.698 | 2.902 | 5.455 | 5.732 |
| 22 | 0.547 | 0.884 | 2.573 | 3.320 | 5.322 | 5.752 |
| 21 | 0.557 | 0.873 | 2.524 | 3.184 | 5.693 | 5.825 |
| 20 | 0.552 | 0.884 | 2.480 | 3.130 | 5.483 | 5.771 |
| 19 | 0.557 | 0.884 | 2.466 | 3.130 | 5.387 | 5.637 |
| 18 | 0.576 | 0.879 | 2.466 | 3.135 | 5.435 | 5.688 |
| 17 | 0.542 | 0.879 | 2.471 | 3.140 | 5.304 | 5.573 |
| 16 | 0.557 | 0.879 | 2.461 | 3.110 | 5.273 | 5.469 |
| 15 | 0.547 | 0.879 | 2.456 | 3.120 | 5.212 | 5.691 |
| 14 | 0.557 | 0.874 | 2.456 | 3.062 | 5.811 | 5.965 |
| 13 | 0.562 | 0.861 | 2.451 | 3.066 | 5.835 | 6.064 |
| 12 | 0.562 | 0.879 | 2.451 | 3.042 | 5.735 | 5.938 |
| 11 | 0.562 | 0.879 | 2.437 | 3.018 | 5.586 | 5.869 |

**Table 1.** *Cont.*

| Floor | First Order | | Second Order | | Third Order | |
|---|---|---|---|---|---|---|
| | NS | EW | NS | EW | NS | EW |
| 10 | 0.552 | 0.890 | 2.471 | 3.047 | 5.654 | 5.864 |
| 9 | 0.562 | 0.859 | 2.427 | 3.018 | 5.659 | 5.811 |
| 8 | 0.552 | 0.874 | 2.441 | 3.022 | 5.522 | 5.820 |
| 7 | 0.543 | 0.879 | 2.422 | 2.964 | 5.420 | 5.786 |
| 6 | 0.555 | 0.830 | 2.422 | 2.939 | 5.405 | 5.659 |
| 5 | 0.544 | 0.879 | 2.417 | 2.979 | 5.391 | 5.698 |
| Mean | 0.554 | 0.877 | 2.461 | 3.051 | 5.487 | 5.765 |

After the natural vibration frequencies were captured, the mode shapes of the structure were determined as described above. The peak point value of the Fourier amplitude spectra at each natural frequency was taken as the amplitude of the mode shape. The phase can be determined according to the cross-correlation function $R_{xy}$ between the test points [45]. If $R_{xy} > 0$, the test points x and y are in the same phase; if $R_{xy} < 0$, this means that the phases of points x and y are opposite. If the phases are the same, the mode shapes have the same sign, otherwise the signs are opposite. Due to data acquisition errors and noise interference, the amplitudes of individual test points in some mode shapes may deviate from the actual situation. In order to better describe the mode shapes of the structure, the normalized amplitudes were averaged in this paper, and the first three mode shapes of the structure in two directions were obtained, as shown in Figure 7. It can be seen that due to the change of the masonry blocks used on the 12th to 15th floors, some mode shapes of the structure have a certain mutation at these positions.

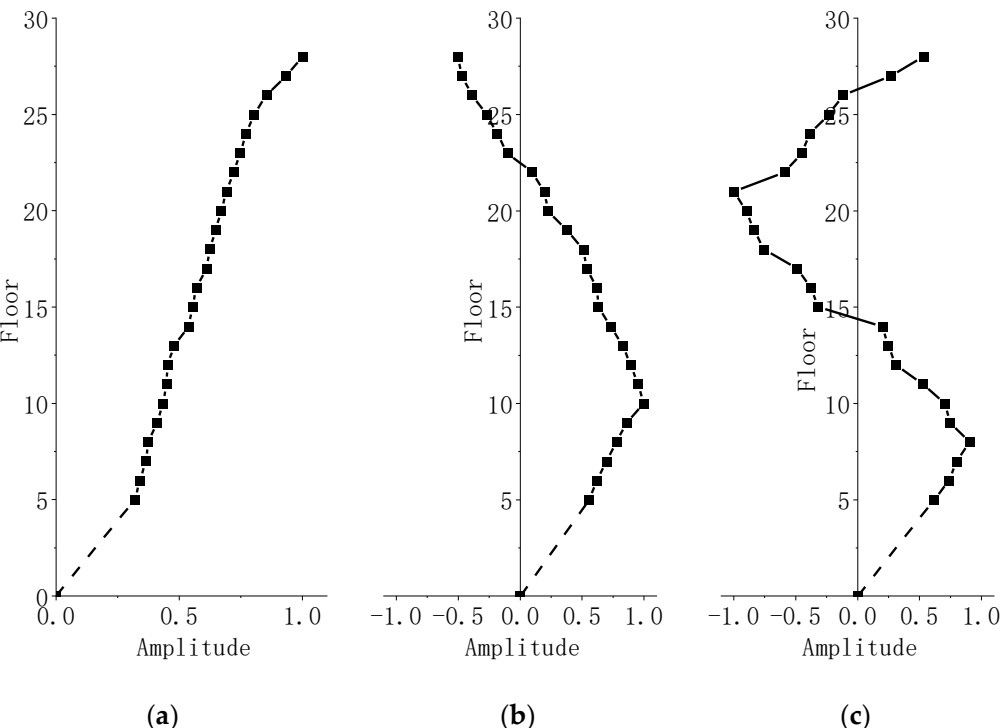

**(a)** **(b)** **(c)**

**Figure 7.** *Cont.*

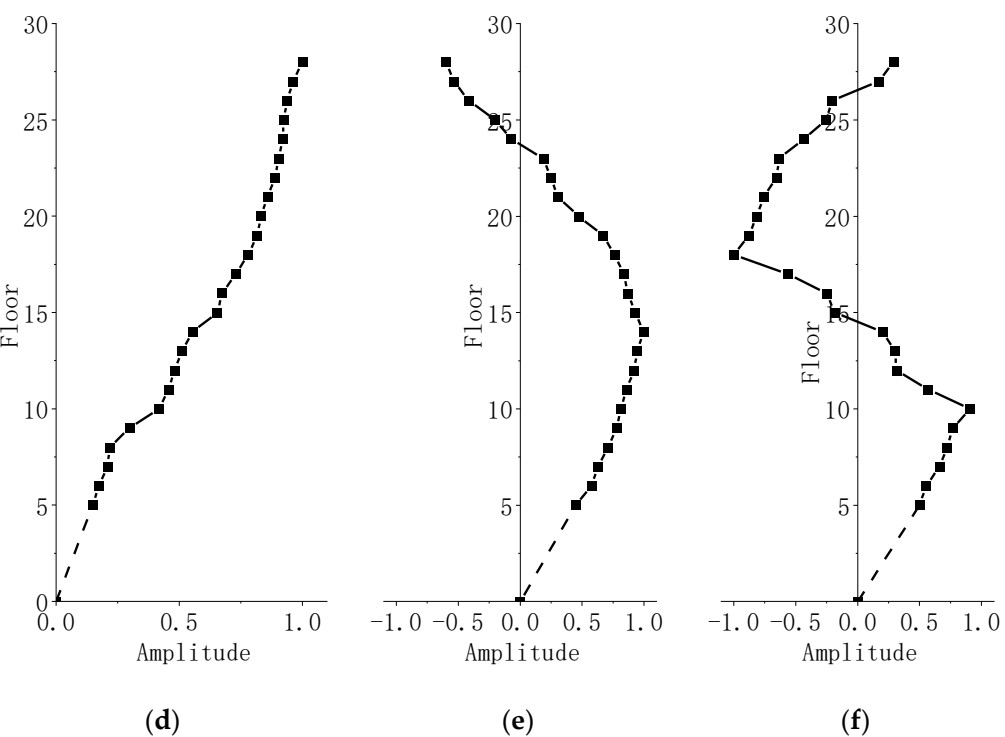

**(d)** **(e)** **(f)**

**Figure 7.** The identified mode shapes of the structure. (**a**) first-order in NS, (**b**) second-order in NS, (**c**) third-order in NS, (**d**) first-order in EW, (**e**) second-order in EW, (**f**) third-order in EW.

### 4.3. Numerical Modal Analysis

In this section, a numerical modal analysis of the reinforced masonry structure was carried out, and a finite element model was established using the finite element software SAP2000 [46]. The interaction between the soil and the structure and the influence of the basement on the main structure were not considered in the modeling. The reinforced masonry wall can be regarded as a shear wall and was simulated using a nonlinear layered shell element. The shell element was divided into concrete and reinforcement layers, and different layers were set with their respective material properties and shell thicknesses. During the program calculation, the strain and stress values at the center of each layer were calculated according to the flat section assumption, and then the internal force of the section was obtained by integrating over the whole section. The nonlinear behavior of reinforced concrete beams was simulated by specifying concentrated plastic hinges at the appropriate locations of the beams. In this paper, only the plastic hinge of the ring beam needed to be defined, and the beam hinge was defined as an M3 moment hinge. Since the transverse shear of the floor slab has little influence on the deformation of the structure, the thin shell element was used for the floor slab in the modeling, and the floor slab was set as a rigid floor slab. For the concrete material, the Mander model and Takeda hysteretic model were used for the constitutive model. In the Mander constitutive model, the restraint effect of the hoop reinforcement was considered, which can comprehensively take into account the influence of the hoop arrangement, spacing, yield strength, and the relative area of the core concrete on the mechanical properties. For the reinforcement material, the Park tri-linear model was used for the stress–strain relation, in which the mechanical characteristics, whereby the stiffness decreases with the increase of strain, can be adequately considered. Moreover, the Kinematical model was adopted for hysteretic behavior. Finally, the finite element modeling for the high-rise reinforced masonry structure is shown in Figure 8.

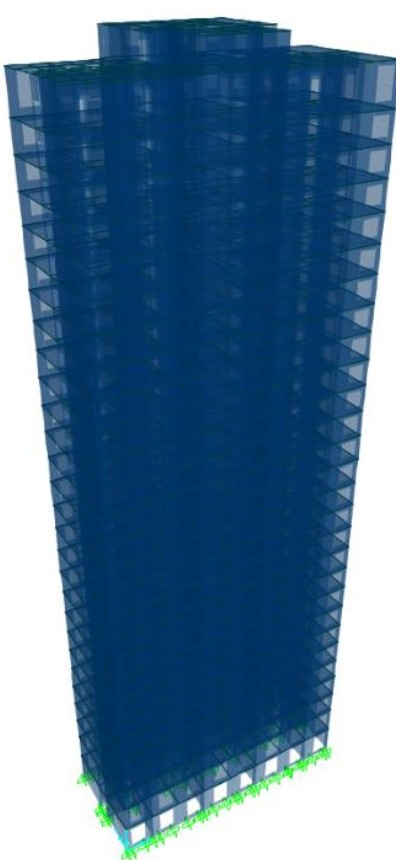

**Figure 8.** Finite element modeling for the high-rise reinforced masonry structure.

The modal analysis of the finite element modeling was carried out, and the first three orders of frequencies in the NS and EW directions obtained from the analysis were compared with the frequency results obtained from the field test using the environmental excitation method, as shown in Table 2. It can be seen that the numerical analysis results were close to the test results, and the maximum relative error was 11.2%, which occurred in the first order of the EW direction.

**Table 2.** The first three-order frequencies of the test and numerical results (Unit: Hz).

|  | First-Order | | Second-Order | | Third-Order | |
|---|---|---|---|---|---|---|
|  | **NS** | **EW** | **NS** | **EW** | **NS** | **EW** |
| Test | 0.554 | 0.877 | 2.461 | 3.051 | 5.487 | 5.765 |
| Numerical | 0.570 | 0.975 | 2.288 | 2.924 | 5.332 | 5.683 |
| Error (%) | 2.9 | 11.2 | −7.0 | −4.2 | −2.8 | −1.4 |

Furthermore, the first three orders of the mode shapes of the structure in both directions obtained from the numerical analysis are given in Figure 9. It can be seen that the mode shape pattern of the structure was basically consistent with the test results.

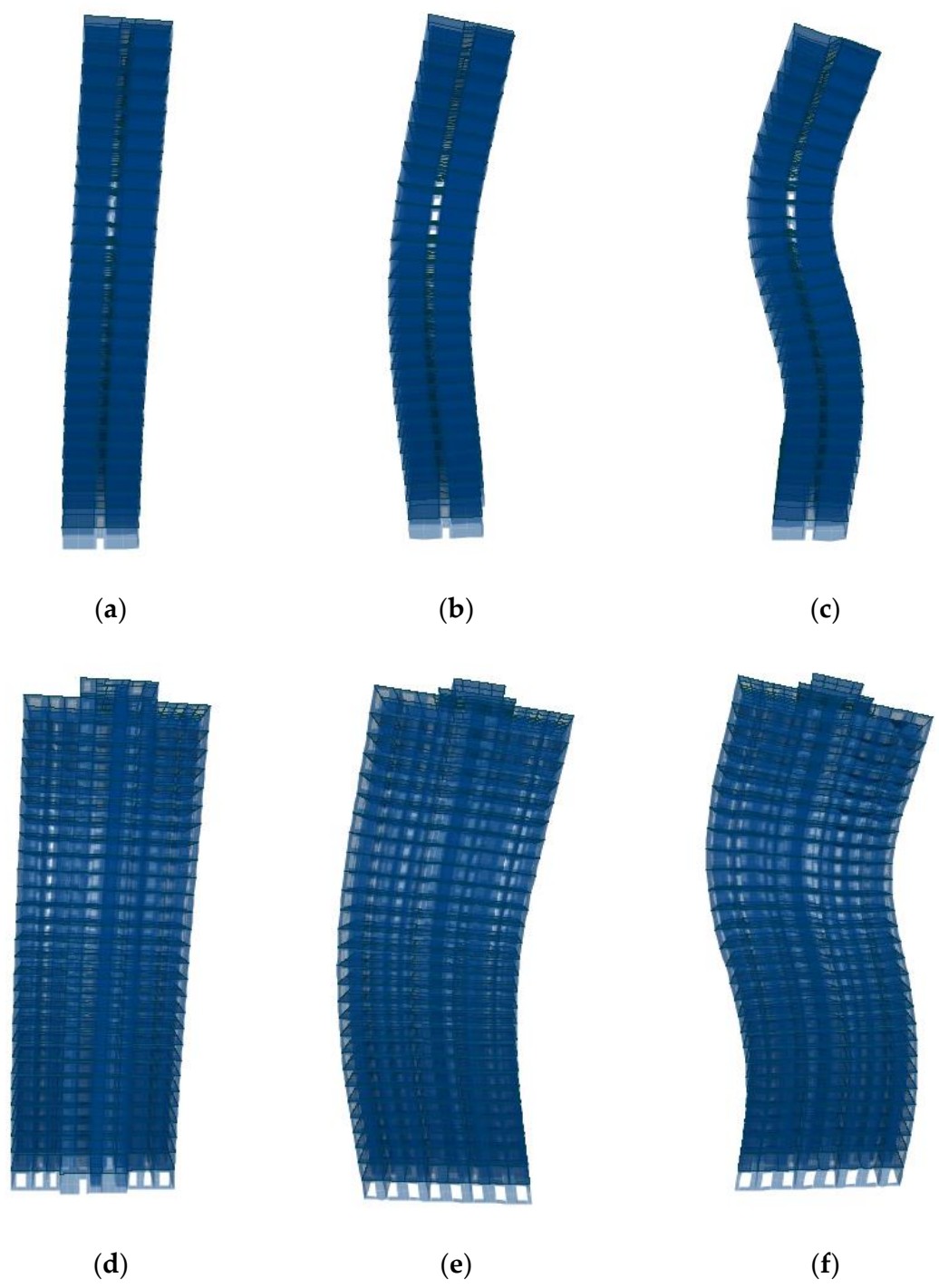

**Figure 9.** The mode shapes of the structures from the numerical analysis. (**a**) first-order in NS, (**b**) second-order in NS, (**c**) third-order in NS, (**d**) first-order in EW, (**e**) second-order in EW, (**f**) third-order in EW.

In order to accurately compare with the test data, the mode shapes of the numerical and test results were further plotted in the form of dotted lines, as shown in Figure 10.

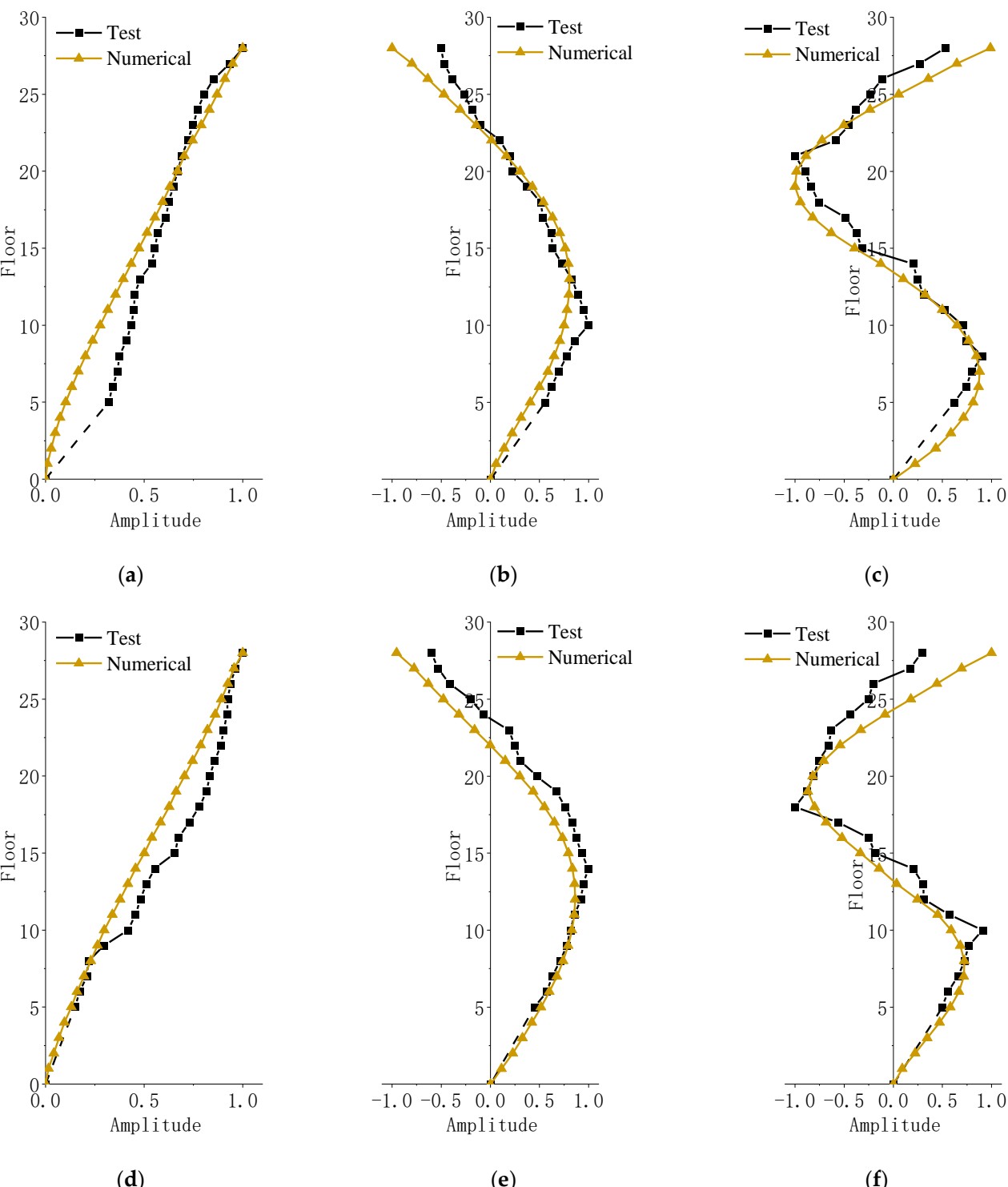

**Figure 10.** Comparison of the mode shapes for the numerical analysis and test results. (**a**) first-order in NS, (**b**) second-order in NS, (**c**) third-order in NS, (**d**) first-order in EW, (**e**) second-order in EW, (**f**) third-order in EW.

It can be seen that the vibration trends of the numerical analysis and the test results were consistent, and the errors were within a reasonable range; thus, the accuracy of the finite element modeling and the test results were mutually verified.

## 5. Structural Vibration Analysis during Songyuan Earthquake

Songyuan City is located on the longest Tan-Lu fault zone in eastern China. The eastern part of this zone is influenced by the trend of the Pacific plate squeezing westward, and the western part is affected by the trend of the Eurasian plate escaping eastward, which has caused Songyuan city to be in a sandwiched state between the two sides. Moreover, due to the excessive geological activities of human beings, the crustal movement may also have been affected to a certain extent. As a result, Songyuan City has experienced frequent earthquakes in recent years. From 31 October to 24 November 2013, there were five earthquakes with a magnitude of five or higher, and which were sensed in Harbin City. Therefore, on the highest reinforced masonry structure in China, a structural seismic response observation array was jointly set up by the Institute of Engineering Mechanics, China Earthquake Administration, the School of Civil Engineering, Harbin Institute of Technology, and Heilongjiang Construction Group.

On 15 August 2017, a magnitude 4.5 earthquake struck Songyuan City. The structural response array was 141.83 km away from the epicenter, and three-component records of strong motion observation were obtained on the 28th floor of the structure. This structural response array record is the first seismic response observation record of a high-rise reinforced masonry structure in China, which provides data support for the study of the seismic performance of ultra-high typically reinforced masonry structures.

After baseline initialization and band-pass filtering using a Butterworth filter of 0.1–35 Hz, horizontal acceleration time history curves were produced, as shown in Figure 11. The peak acceleration values in the NS and EW directions were 4.2 cm/s$^2$ and 2.8 cm/s$^2$, respectively. It can be seen that the peak values of the acceleration time history curves were small, since the structural response array was far away from the epicenter and the magnitude of the earthquake was relatively low.

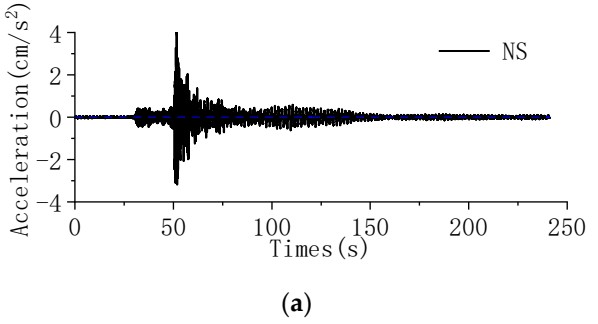
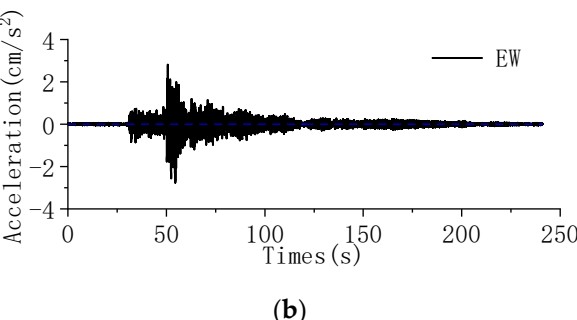

(**a**)

(**b**)

**Figure 11.** Acceleration time history curves of the structural response records during Songyuan earthquake (**a**) NS; (**b**) EW.

The Fourier amplitude spectra of structure in the NS and EW directions for the processed data are shown in Figure 12. The first three-order natural vibration frequency values for each horizontal direction were identified using the peak-picking method and compared with the environmental vibration data analysis results, as shown in Table 3.

It can be seen that the first three-order natural vibration frequency values calculated using the acceleration response records were close to the environmental vibration data analysis results. The maximum absolute error occurred in the third order frequency in the EW direction, which was −0.373 Hz, and the relative errors were all within 6.5%. Due to the fact that only one set of records was obtained during this earthquake, occasional errors occurred, in recognition of the small sample amount of calculation.

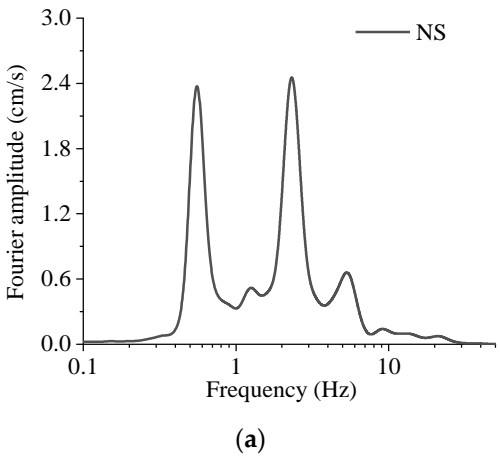 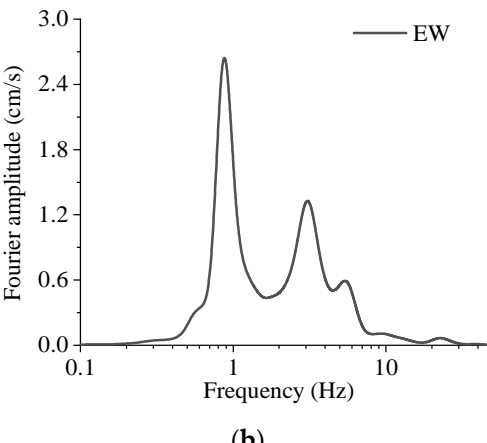

**Figure 12.** Fourier amplitude spectra of the structure in the NS and EW directions for structural response records (**a**) NS; (**b**) EW.

**Table 3.** Natural vibration frequency values obtained from the structural response records (Unit: Hz).

|  | First Order | | Second Order | | Third Order | |
| --- | --- | --- | --- | --- | --- | --- |
|  | **NS** | **EW** | **NS** | **EW** | **NS** | **EW** |
| Earthquake (Hz) | 0.555 | 0.876 | 2.319 | 3.067 | 5.322 | 5.392 |
| Environmental vibration data (Hz) | 0.554 | 0.877 | 2.461 | 3.051 | 5.487 | 5.765 |
| Absolute error (Hz) | 0.001 | −0.001 | −0.142 | 0.016 | −0.165 | −0.373 |
| Relative error (%) | 0.18 | −0.11 | −5.77 | 0.52 | −3.01 | −6.47 |

Furthermore, the seismic response records captured during this earthquake have sufficiently long data. In the acceleration time history, the data recorded before and after the earthquake were actually equivalent to the environmental vibration data of the structural response. In order to analyze whether the natural vibration frequency of the structure changed before and after the earthquake, the first 20 s and last 30 s of the structural response data were intercepted as environmental vibration data, which were used to identify the natural vibration frequency before and after the earthquake. The Fourier amplitude spectra in the NS and EW directions are shown in Figure 13, and first three-order natural vibration frequency values in the two directions before, during, and after the earthquake are listed in Table 4.

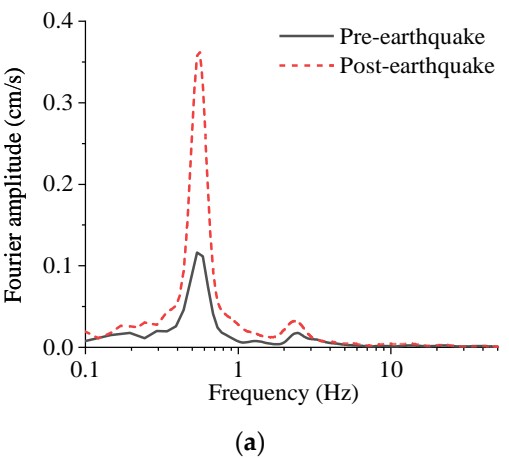 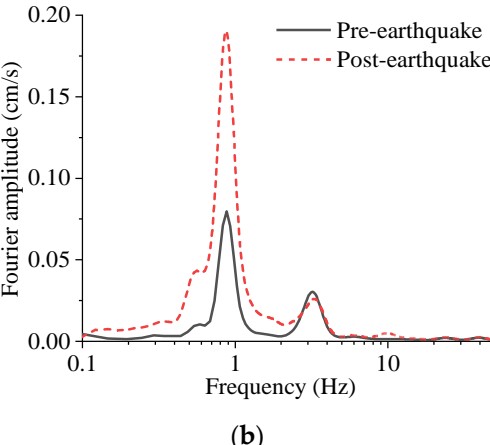

**Figure 13.** Fourier amplitude spectra in the NS and EW directions before and after the earthquake (**a**) NS; (**b**) EW.

**Table 4.** Natural vibration frequency values before, during, and after the earthquake (Unit: Hz).

| | First Order | | Second Order | | Third Order | |
|---|---|---|---|---|---|---|
| | NS | EW | NS | EW | NS | EW |
| Pre-Earthquake | 0.537 | 0.879 | 2.441 | 3.223 | 5.273 | 5.908 |
| Earthquake | 0.555 | 0.876 | 2.319 | 3.067 | 5.322 | 5.392 |
| Post-Earthquake | 0.562 | 0.879 | 2.344 | 3.296 | 5.249 | 5.908 |

It can be seen that the difference among the first three-order natural vibration frequencies before, during, and after the earthquake was very small, and the maximum change rate was only 8.7%, which appeared at the third-order frequency. In combination with the peak response recorded during this earthquake, it can be judged that the seismic performance of the structure did not change significantly, the structure was basically undamaged, and it remained in an elastic state.

## 6. Conclusions

In this paper, environmental vibration data obtained from a field test of a high-rise reinforced masonry structure in Harbin and the structural response record obtained from an earthquake were processed and analyzed. The peak-picking method was used to identify the dynamic characteristics of the structure, and the conclusions were as follows:

(1) With the increase of floors, the Fourier amplitude spectra of the structural environmental vibration data contained more spectral information, and the frequencies at the peak points for each segment of data were also more consistent, which could better calculate the natural vibration frequency of the structure. Specifically, the first three-order frequencies in the NS direction were 0.554, 2.461, and 5.487 Hz, respectively, and the first three-order frequencies in the EW direction were 0.877, 3.051, and 5.765 , respectively. Then, the mode shapes of the structure at the corresponding natural frequencies were determined using the ratio of the amplitudes of the different measuring points. Furthermore, comparing the numerical modal analysis results with the test results revealed the accuracy of the obtained structural dynamic characteristics.

(2) Based on the structural response records during an earthquake in Songyuan, the first three-order natural frequencies of the structure in the NS and EW directions were identified, which were 0.555 Hz, 2.319 Hz, and 5.322 Hz, and 0.876, 3.067, and 5.392 Hz, respectively, and basically the same as the results obtained from the environmental vibration data. The structural response data before, during, and after the earthquake were then analyzed. It can be seen that the natural frequency of the structure changed very little, and the maximum relative error was only 8.7%. Considering the small amplitude of the seismic response records, it could be determined that the structure remained in an elastic state after the earthquake.

(3) Based on the comprehensive analysis results, during modal parameter identification of high-rise and regular-plan structures, and under the premise of cost and time limits, it is suggested that a structural response array is set up on the floors above one-third of the height of the structure, besides the first floor and the free field, from which better identification results and the ideal mode shapes can be obtained.

**Author Contributions:** Conceptualization, B.Z., X.W. and J.K.; methodology, B.L. and J.K.; software, B.L.; validation, B.Z., X.W. and J.K.; formal analysis, B.L. and J.K.; investigation, B.Z. and C.Z.; resources, B.Z. and J.K.; data curation, B.Z. and C.Z.; writing—original draft preparation, X.W. and B.L.; writing—review and editing, B.Z., X.W. and C.Z.; visualization, B.L.; supervision, J.K.; project administration, C.Z.; funding acquisition, B.Z. All authors have read and agreed to the published version of the manuscript.

**Funding:** This research was funded by Scientific Research Fund of Institute of Engineering Mechanics, China Earthquake Administration (2021EEEVL0301), the National Key Research and Development

Program of China (2018YFE0109800, 2017YFC1500601), and the Heilongjiang postdoctoral fund in China (LBHQ18032). This support is greatly appreciated.

**Institutional Review Board Statement:** Not applicable.

**Informed Consent Statement:** Not applicable.

**Acknowledgments:** Grateful acknowledgment is given to Ruizhi Wen, Maosheng Gong, Qiang Ma, and Yefei Ren at the Institute of Engineering Mechanics, China Earthquake Administration, and Changhai Zhai, Fenglai Wang at School of Civil Engineering, Harbin Institute of Technology, for their guidance and suggestions on the theoretical analyses.

**Conflicts of Interest:** The authors declare no conflict of interest.

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
