# Peer review of "Natural Vibration Characteristics Analysis of a High-Rise Reinforced Masonry Structure Based on Field Test Data"

_buildings, doi:10.3390/buildings12091457_

Round 1
Reviewer 1 Report
This paper described a structural dynamic characteristics identification method for determining mode shapes and vibration frequencies of a high-rise buildings. My comments on the paper are listed below.
1. The floor locations and plan arrangements of sensors are missing.
2. The environmental excitation is small and influence of noise should be very high. The results would not be accurate.
3. The torsional modes have not been considered.
4. If only the peak response were identified, please clarify how to determine the displacement direction.
5. If the peak-picking method is not new, what is the academic contribution of this paper?
Reviewer 2 Report
The article addresses an important and very interesting topic of the natural vibration characteristics analysis of a high-rise reinforced masonry structure based on field test data, which is appreciated. The study includes the experimental and numerical research. The purpose of this paper was identifying of the natural vibration frequency and mode shape of the structure from the field vibration test on the highest reinforced masonry structure in China. At the same time, a group of structural response records captured in Songyuan earthquake was processed and analysed. In addition, the health status of the structure was judged by the change of natural vibration frequency information of the structure (before, during and after the earthquake). The Reviewer has some concerns regarding to the all paper, because in this paper the Reviewer cannot find the numerical model with their properties. Based on the table 1 it can be concluded that this paper should be included the numerical analysis. In addition, the description of vibration test was not accurate, that we cannot compare the results from this paper to the other papers (results). Thus, the correct validation of results is unlikely. Furthermore, in the Figure 7 the Reviewer cannot see the mode of shape (probably these results are from modal analysis). Also, the Figure 6 is not clear (what are mean the line on the charts?). Generally, in this paper the English language is good, but some sentence should be more clearly (too long, some words are not true e.g. “pulsation”, “array” etc.), thus please check the all text of Native Speaker. In opinion of Reviewer this paper should be rejected.
Finally, I hope that my comments will be helpful for the authors.
Reviewer 3 Report
see the file attached

Round 2
Reviewer 1 Report
My previous comments have been addressed. I have no further comments.
Author Response
Thanks to the reviewer for the valuable suggestions and comments.
Reviewer 2 Report
Thank you for your improving. The Reviewer have still concern to the verification of the results.
In general, this paper should be rejected, but if other reviewers have a different opinion, so I tried to give Authors a second chance to improve their answers. In my opinion, their response of these question
Reviewer: “The Reviewer has some concerns regarding to all paper, because in this paper the Reviewer cannot find the numerical model with their properties. Based on the table 1 it can be concluded that this paper should be included the numerical analysis. In addition, the description of vibration test was not accurate, that we cannot compare the results from this paper to the other papers (results). Thus, the correct validation of results is unlikely.”
ae off-topic answers
Authors: “We sincerely appreciate reviewer’s valuable comments. In practical research, theoretical analysis, experimental test and numerical simulation are the most common methods to study the seismic performance of structures. All these analyses make different degrees of simplifying assumptions on the used models, and the reliability of the model can't be well verified due to the lack of the test response data of structures. In this paper, the field test is performed on a reinforced masonry structure in China. The nutural vibration frequencies and the mode shapes of the structure are obtained by using the test data, which can be further used to verify the rationality of numerical simulations. In the field test, high-sensitivity strong-seismometer is adopted, the seismometer was set up on all floors, and the field test was performed at night to prevent the interference of external environment. These guarantee the accuracy of the recorded data. Furthermore, each record in the original data is recorded for a long time, and each piece of data is divided into 10 segments of records. With these data, the nutural vibration frequencies obtained from the 10 segments of data of the same floor and the data of different floors are compared, and these values are generally consistent, which indicates the accuracy of the results tested in this paper.”
In the summary, Authors did not dispel my doubts. Thus, maybe while reading good exemplary papers:
• https://doi.org/10.3390/ijerph19095441
• https://doi.org/10.1016/j.jsv.2014.04.052
• https://doi.org/10.7307/ptt.v31i3.3000
can be able to answer of my questions. Finally, I hope that my comments will be helpful to the Authors.
Round 3
Reviewer 2 Report
Dear Authors, thank you for your improving. The section 4.3, which was add is very important and have significant contribution in your paper. In current version of this paper can I recommend to the publish.